# Food and Gut Microbiota-Derived Metabolites in Nonalcoholic Fatty Liver Disease

**DOI:** 10.3390/foods11172703

**Published:** 2022-09-05

**Authors:** Min Kyo Jeong, Byeong Hyun Min, Ye Rin Choi, Ji Ye Hyun, Hee Jin Park, Jung A Eom, Sung Min Won, Jin Ju Jeong, Ki Kwang Oh, Haripriya Gupta, Raja Ganesan, Satya Priya Sharma, Sang Jun Yoon, Mi Ran Choi, Dong Joon Kim, Ki Tae Suk

**Affiliations:** Institute for Liver and Digestive Diseases, Hallym University College of Medicine, Chuncheon 24252, Korea

**Keywords:** gut microbiota, nutrient, short-chain fatty acid, gut–liver axis, metabolites

## Abstract

Diet and lifestyle are crucial factors that influence the susceptibility of humans to nonalcoholic fatty liver disease (NAFLD). Personalized diet patterns chronically affect the composition and activity of microbiota in the human gut; consequently, nutrition-related dysbiosis exacerbates NAFLD via the gut–liver axis. Recent advances in diagnostic technology for gut microbes and microbiota-derived metabolites have led to advances in the diagnosis, treatment, and prognosis of NAFLD. Microbiota-derived metabolites, including tryptophan, short-chain fatty acid, fat, fructose, or bile acid, regulate the pathophysiology of NAFLD. The microbiota metabolize nutrients, and metabolites are closely related to the development of NAFLD. In this review, we discuss the influence of nutrients, gut microbes, their corresponding metabolites, and metabolism in the pathogenesis of NAFLD.

## 1. Introduction

Genetic background, environmental factors, lifestyle, host physiology, and dietary intake affect human health and the risk of disease throughout life. Of these components, food and nutrients play an essential role in maintaining health [1,2]. Dietary nutrients are absorbed in the intestine and are involved in preserving health through various actions. For example, a high calorie diet may cause metabolic syndrome, and the only approved treatment is exercise and diet control.

Microbiota is a collective term for the microorganisms that live in or on the human body. The microbiome refers to the collection of genomes from all the microorganisms in the environment. In the intestine, nutrients are metabolized by gut microbiota, and the metabolized nutrients and their metabolites move to the liver through the portal vein and are involved in various metabolic processes. This gut–liver axis is involved in the metabolism of nutrients and microbiota in the gut–liver axis play an important role. Trillions of microbiota are present in the intestine and some microbiota may contribute to the pathogenesis of liver diseases [3]. The gut microbiota affect liver disease through various mechanisms, including chronic systemic inflammation, increased gut permeability, the production of short-chain fatty acids, and changes in the metabolism [4]. Dysbiosis is a condition in which microbiome balance is disrupted, resulting in an imbalance in the composition of the microbiota. Gut dysbiosis is related with NAFLD progression by increasing the translocation of microbial metabolites into the liver [5,6].

The human gastrointestinal tract contains many microbiota composed of bacteria, viruses, and fungi. It provides a platform for multiple interactions between the host and the gut microbiome [7]. The gut microbiome forms a complex ecosystem that comprises several biological networks and functional mechanisms that interact with one another. Our knowledge about the crucial roles of the gut microbiome in growth, the immune response, inflammation, and metabolism has increased gradually in the past decade [8,9]. Diversity in diet patterns and food intake are the main drivers of the composition of gut microbial clusters, and a diverse and high-quality diet leads to healthy gut microbiota [10]. An uncontrolled diet may cause dysbiosis, and gut-derived microbial lipopolysaccharides (LPS), a toxic component of the bacterial wall, play an important role in the development of NAFLD [6].

Pathologically, NAFLD includes simple steatosis, nonalcoholic steatohepatitis (NASH), and NASH-related cirrhosis [11]. Total energy intake and nutrients are associated with the development of liver disease, which can be characterized by the proper selection of active nutrients that can play a role in the regulation of the nutritional metabolism [12,13]. Several studies have indicated that an inappropriate diet may lead to a fatty liver [14]. To date, recent research has reported that the composition of gut microbes changes according to nutritional status and that their metabolites are associated with the development of NAFLD. In this study, we intend to discuss the relationship between nutrients and the microbiota-derived metabolites in the pathogenesis of NAFLD.

## 2. Food and Gut Microbiota

There are many microbiota in our intestines, which are collectively called the “gut microbiota” [15]. The gut microbiota are comprised of all the organisms that are present in the gastrointestinal tract. Bacteria, viruses, fungi, and protozoa make up approximately 100 trillion microbiota in the human gastrointestinal tract [16]. Each microbial species can form colonies from 10^12^~10^14^ cells/mL [17]. The composition and function of the gut microbiota can be affected by various factors, including age, gender, genetic factors, lifestyle, medication, and dietary differences [18]. Environmental and genetic factors, as well as changes in the gut microbiota that play a role in the pathogenesis of metabolic disorders, are also implicated. Most gut bacteria exist in a commensal form, including *lactobacilli* and *bifidobacteria*. However, there are also *Enterococcus* and *Escherichia coli*, which can be harmful under certain conditions such as dysbiosis or in various diseases [19,20]. 

The gut microbiome contributes to nutrient processing and signaling and produces metabolites with essential functions, such as tryptophan, bile acid, choline, and short-chain fatty acid (SCFA) metabolisms [21]. Gut dysbiosis is associated with various diseases including obesity, insulin resistance, coronary heart disease, gut disease, metabolic syndrome, and infection [22,23]. 

Recent advances in immunology and microbiology have supported new hypotheses. First, the composition of gut microbiota clusters was significantly affected by diet patterns [24]. Each individual harbors his own distinctive pattern of microbial composition, and changes in the diet lead to changes in the composition (Table 1). Second, it has been found that the typical microbial composition and microbiota-derived products found in various laboratories have shown unexpected effects on immune and inflammatory responses [25].

## 3. Nonalcoholic Fatty Liver Disease

NAFLD is recognized as the most common disease among chronic liver diseases, and the prevalence of NAFLD has increased recently in the Asia-Pacific region [41]. Diabetes and metabolic syndrome are on the rise worldwide, and as a result, the prevalence of NAFLD is increasing. NAFLD causes NASH, along with other persistent liver diseases, over time, which can subsequently lead to cirrhosis and liver cancer [42,43]. One of the main causes of NAFLD is an inadequate diet, along with an oversupply of calories and an unbalanced intake of fats, grains, fruits, vegetables, protein, and ω-3 fatty acids at the same time [13]. NAFLD is closely related with changes in gut microbiota composition and metabolic activity (Table 2) [44].

In the “multi-hit” theory, various factors, such as dietary intake, environmental factors, insulin resistance, adipose tissue dysfunction, alterations in the gut microbiota, and genetic predisposition, are involved in the pathophysiology of NAFLD [55]. Gut dysbiosis increases the gut permeability of bacteria and increases exposure to harmful substances that increase inflammation and fibrosis in the liver [56]. Lifestyle changes are an effective treatment for NAFLD, leading to weight loss and improving metabolic diseases including heart disease and endocrine disease [57]. Therefore, it is of practical significance to study the relationship between dietary intake-induced liver lipid accumulation and nutrient intake to prevent diseases associated with NAFLD [58]. 

The gut–liver axis is important for the development, progression, and prognosis of NAFLD. According to the kind of diet and food, gut dysbiosis can occur, and bacteria-derived substances (endotoxin, bacterial DNA, microbe-associated molecular patterns (MAMPs), damage-associated molecular patterns (DAMPs), pathogen-associated molecular patterns (PAMPs), and nematode-associated molecular patterns (NAMPs)) directly affect the liver through the portal vein [6]. In addition, alcohol-secreting strains cause an inflammatory response in the liver and body [59].

## 4. Nutrients Associated with Microbiota in Nonalcoholic Fatty Liver Disease

Food-derived metabolites and metabolites associated with the gut microbiota are described in Table 3.

### 4.1. Bile Acid

Bile acids (BAs) are a major component of bile juice and play a crucial role in the absorption of dietary fat and some vitamins because they are bipolar [70]. It is known that bile acids are involved in digestion as surfactants. In many studies in recent decades, bile acids have been reported to act as a wide range of signaling hormones in the human body by binding to various receptors in the cell membrane or nucleus, thus regulating fat metabolism, glucose metabolism, inflammation, and the growth of gut microbes [71,72,73]. 

BAs are generally divided into primary and secondary BAs. Primary BAs are synthesized directly from cholesterol, such as cholic acid (CA) and chenodeoxycholic acid (CDCA). Most primary BAs are synthesized through the cytochrome p450 enzyme, and among them, cholesterol 7a-hydroxylase (CYP7a1) is a key factor in regulating the synthesis of bile acids [74,75]. Secondary BAs secrete bile into the duodenum during digestion, move along the intestines, and are produced by the metabolism of gut microbes. At this time, deoxycholic acid (DCA) is produced from CA, chenodeoxycholic acid is produced from CACA, and various BAs are present [76,77]. 

The role of BAs in NAFLD/NASH is increasingly acknowledged [78]. The risk of liver injury is increased by the dysregulated BA metabolism in NAFLD patients [79]. Thus, increased bile acid exposure may be involved in hepatic injury and the pathogenesis of NAFLD and NASH [80]. 

The complex mechanism of BA action in the development of NAFLD/NASH is controlled primarily by the farnesoid X receptor (FXR), activated by primary BAs, and Takeda G protein-coupled receptors 5 (TGR5), activated by secondary BAs [81]. Since bile acids are cytotoxic, they can cause liver damage and hepatitis, so it is important to maintain a proper amount within the cells. FXR performs various functions related to fat and glucose metabolism and is also involved in the regulation of inflammatory responses. In addition, FXR plays a critical role in maintaining metabolic homeostasis by regulating the synthesis, secretion, and transport of bile acids [82,83,84]. Many recent studies have provided strong evidence that FXR activity inhibits lipopolysaccharide (LPS)-stimulated NF-κB activation and signaling, thereby suppressing hepatic inflammation and fibrosis, and resulting in a hepatoprotective effect [85]. However, one study found that regulating gut microbiota to inhibit gut FXR resulted in BA alteration and increased BA metabolites, further contributing to the amelioration of rat liver steatosis. [86]. Therefore, FXR regulates the gut microbiota and has a wide range of effects in the treatment of NAFLD.

### 4.2. Short-Chain Fatty Acid

SCFAs are generated through the fermentation of indigestible polysaccharides such as dietary fiber by gut microbes in the large bowel [87]. This fermentation supports the growth of SCFAs and specialized microbiota that produce gases, and the main metabolites of the SCFAs are propionate, acetate, and butyrate [88]. In general, propionate affects hepatic lipogenesis and gluconeogenesis, whereas peripheral acetate functions as a substrate for cholesterol synthesis [89]. Ninety-five percent of SCFAs produced in the cecum and large bowel are rapidly absorbed by colonic cells, and the remaining 5% are excreted in the feces [90]. Additionally, SCFAs modulate the production of several inflammatory cytokines, including TNF-α, IL-6, IL-2, and IL-10 [91]. 

Butyrate activates gut gluconeogenesis, which has controlling effects on glucose and energy homeostasis [92]. Butyrate is necessary for epithelial cells to consume large amounts of oxygen by beta-oxidation, creating a hypoxic state that maintains oxygen balance in the intestine and prevents dysbiosis in the gut flora [93]. Butyrate also regulates diet-induced insulin resistance in animals. One study reported that sodium butyrate prevents steatohepatitis by modulating immune responses in the gut and liver and reducing microbial imbalance and endotoxemia in high-fat diet-induced mice [94]. As a result, SCFAs can prevent the progression of fatty liver to NASH by promoting hepatic glucagon-like peptide (GLP)-1 receptor expression [68]. Furthermore, increased GLP-2 levels by microbial or subcutaneous GLP-2 administration to mice reduces gut permeability, as well as plasma LPS and cytokine levels, resulting in reduced hepatic oxidative stress and inflammation [69]. Activation of G-protein-coupled receptor-43 revealed the antiinflammatory effect of SCFAs, and colitis was ameliorated by acetate supplementation. This is a finding not found in G-protein-coupled receptor-43 knockout mice [60]. Therefore, SCFAs improve gut permeability and prevent the delivery of harmful microbiota-derived substances and metabolites to the liver. 

### 4.3. Amino Acid and Tryptophan

Amino acids are major building blocks of cellular proteins and precursors of nitrogenous substance synthesis. Polyamines, nitric oxide, catecholamines, creatine, and dopamine are typical amino acids and are essential in systemic homeostasis [95]. Although most dietary proteins are digested and absorbed in the small intestine, significant amounts of proteins and amino acids reach the colon, where various related bacteria degrade them [96,97]. 

Bacterial metabolites arising from tryptophan, phenylalanine, and tyrosine are known to be involved in the pathogenesis of NAFLD. These amino acids and their metabolites from the gut microbiota have been shown to have a variety of effects on the liver [98]. Tryptophan is an essential amino acid for humans and is found in many protein-based foods, including milk, meat, fruits, and seeds [99].

Dietary tryptophan catabolites include indole, tryptamine, indole ethanol, indole propionic acid, indoleacetic acid, indole lactic acid, skatole, indole aldehyde, and indole acrylic acid, and they may affect host physiology in numerous ways [100]. According to recent evidence, the metabolites of indole are known to be effective in gut and hepatic protection. As such, tryptophan availability is a key factor in controlling protein biosynthesis. This may be one of the important reasons why the immune system uses tryptophan to limit the proliferation of pathogens and malignant cells [101]. 

Indole is a beneficial signal for maintaining the gut barrier by increasing the expression of antiinflammatory cytokine interleukin (IL)-10 and inhibiting the tumor necrosis factor (TNF)-α-mediated activation of nuclear factor kappa-light-chain-enhancer of activated B cells (NF-κB) and proinflammatory chemokine IL-8 expression [102]. Indoleamine 2,3-dioxygenase (IDO), the most undigested tryptophan, is absorbed and converted to kynurenine through the regulation of a key rate-limiting enzyme expressed in the gastrointestinal tract [103]. A study of IDO found that high-fat diet-induced IDO-/-mice were less infiltrated with inflammatory macrophages and were protected from obesity-related fatty liver and insulin resistance [104]. Therefore, the gut microbiota might regulate IDO activity to affect the development of NAFLD. 

Indole compounds strengthen the tight junctions of epithelial cells and relieve the gut’s inflammatory response and damage [105]. There is currently firm evidence linking indoles and improvements in gut and liver health. Indole components induce signal transducer and transcriptional activator 3 (STAT3) phosphorylation by stimulating the secretion of IL-22 through the aryl hydrocarbon receptor (AhR), thus promoting the proliferation of gut epithelial cells and restoring the barrier function [106]. The treatment of hepatic and inflammatory severity in mice on an indole-attenuated high-fat diet was associated with the modulation of 6-phosphofructo-2-kinase/fructose-2,6-biphosphatase 3 and the normalization of the gut microbiota [107]. 

Glutamine is a nonessential amino acid and is considered to be an essential nutrient in severe disease, such as cardiovascular disease, liver disease, and intestinal disease [108,109]. As glutamine is one of the abundant amino acids in the body, it regulates protein synthesis and breakdown and plays an important role in regulating acid–base balance, enhancing immune function, and improving adaptability to stress [110]. Glutamine also appears to play an important role in the gut through glutathione production, which may contribute to the prevention of oxidative damage to the gut [111]. 

### 4.4. Choline

Choline is composed of a phospholipid of the cell membrane and is the main nutrient supplied by food and synthesis in the body [112]. Choline is related to biological mechanisms, such as lipid-related pathways and the enterohepatic circulation of bile [112]. Choline-deficient intake is associated with obesity and insulin resistance, and consequently causes NASH through the inhibition of very low-density lipoprotein production [113,114,115,116]. 

Some gut microbiota may selectively metabolize choline to trimethylamine (TMA). TMA is transformed to trimethylamine-N-oxide (TMAO) by flavin-containing mono-oxygenase in the liver [117,118]. Microbiota-related choline metabolism reduces choline bioactivity and creates a choline deficiency, causing metabolic syndrome [119]. In the previous report, transgenic mice models (knockout of a choline-utilizing C) with modified gut microbiota (choline-metabolizing *Escherichia coli* group vs. no choline-metabolizing *E. coli* group). The choline-utilizing group changed the composition of the gut microbiota and metabolites, which induced metabolic disease [119]. In a human trial with 15 subjects, the choline amount in food altered the composition of choline-related gut microbiota, such as *Gammaproteobacteria* and *Erysipelotrichi* [120]. NAFLD patients showed increased levels of TMAO in the serum [121]. TMAO changes the glucose metabolism and increases insulin resistance [122]. Furthermore, TMAO is related to inflammation in adipocyte and insulin resistance by activating proinflammatory cytokine C-C motif chemokine 2 [122]. However, the exact mechanism of TMA in the pathogenesis of NAFLD is not fully demonstrated.

### 4.5. Polyphenol

Polyphenols are naturally occurring bioactive components in fruits and vegetables, and they are the most abundant antioxidants in the human diet [123]. Polyphenols include tea, fruits, vegetables, roots, seeds, cocoa, and wine, which represent a variety of bioactive substances. Approximately 90% to 95% of polyphenols ingested in the diet reach the colon and are then transformed into bioactive products by the gut microbiota [124,125]. In turn, polyphenols metabolized by the gut microbiota improve lipid regulatory bioavailability and act as antioxidants, thus interfering with various metabolic processes that affect cancer cell growth [126,127]. Recent systematic studies demonstrated that polyphenol intake reveals prebiotic properties that control the host’s health by controlling gut dysbiosis and maintaining homeostasis [128,129]. The prebiotic effect of polyphenols is mainly associated with the establishment of probiotics or the inhibition of pathogenic bacteria, resulting in reduced endotoxins, the induction of inflammatory immune responses in the intestine, and other health benefits, such as improved bowel habits and overall well-being [130]. 

Furthermore, various polyphenols have a protective effect on NAFLD by attenuating oxidative stress, insulin metabolism, and inflammation [131]. Surprisingly, polyphenols showed various effects on lipid metabolism, oxidative stress, inflammation, and insulin resistance, which are the most important pathological processes in the pathogenesis of the liver disease [132]. Additionally, nuts have prebiotic properties due to their fatty acid profile and high contents of protein, fiber, phytosterols, vitamins, and phenols [133]. Phenols show multifaceted effects on chronic liver diseases through a well-known mechanism, indicating a potential for the treatment of liver disease [134].

### 4.6. ω-3 Polyunsaturated Fatty Acid

ω-3 polyunsaturated fatty acids (PUFAs) are considered as prebiotics, and the consumption of an ω-3-rich diet has been considered beneficial for health. Fish oil contains more than 40 fatty acids, but sources of ω-3 PUFAs include eicosapentaenoic acid (20:5) and alpha-linolenic acid (18:3) in plant oils and docosahexaenoic acid (22:6) in fish oils [135]. Eicosapentaenoic acid and docosahexaenoic acid are found in significant amounts in fish and other seafood, so they are all marine n-3 fatty acids [136]. The ω-6 PUFAs (γ-linolenic acid (18:3), arachidonic acid (20:4), and linoleic acid (18:2)) and ω-3 PUFAs are well known to attenuate inflammation [137]. As a result, an increase in the dietary intake of ω-6/ω-3 PUFAs has been associated with a higher risk of developing diseases such as obesity, arthritis, diabetes, cardiovascular disease, and liver disease [138,139,140]. Previous studies have shown that ω-3 PUFAs might modulate fatty acid metabolism, thus increasing hepatic fatty acid β-oxidation and antiinflammatory effects in NAFLD [141]. Therefore, more attention has been given to the role of ω-3 PUFAs in NAFLD. 

ω-3 PUFAs directly affect the composition and diversity of gut microbiota. The unbalanced intake of ω-3/ω-6 PUFAs can lead to a significant increase in the *Firmicutes/Bacteroides* ratio and dysbiosis [142]. In addition, fish oil exerts an inhibitory effect on various bacteria, and ω-3 PUFAs can have a beneficial effect on the gut microbiota, which reduces the growth of *Enterobacteria*, increases the growth of *Bifidobacteria*, and increases metabolic endotoxins. It also potentially inhibits hematuria-associated inflammation [28]. Thus, it is suggested that the gut microbiota might regulate the absorption, bioavailability, and biotransformation of ω-3 PUFAs [143]. Finally, ω-3 PUFAs trigger a healthy chain reaction to increase the amounts of SCFA, serve as farnesoid X receptor (FXR) ligands, and create healthy bile acid reservoirs, thus contributing to liver health maintenance and the inhibition of hepatic inflammation [29,144].

### 4.7. Others

Among the heterogeneous groups of plant-derived carbohydrates, dietary fiber is not digested by the host and is selectively fermented by the gut microbiota [145]. A diet with legumes, vegetables, grains, and fruit is usually related to fiber intake [146]. The microbiota ferment soluble fiber to produce SCFAs. SCFAs, which are major energy sources for colon cells, are related to the growth of microbiota, the inhibition of pathogenic bacteria, fat reduction, an increase in insulin sensitivity, and systemic inflammation reduction [147]. Fermented foods including vinegar, olives, yogurt, bread, cheese, butters, wine, beer, pickle, sauerkraut, soy sauce, and yogurt are rich in SCFAs

A vegetable diet, especially abundant in indigestible carbohydrates (fiber), is related to a high proportion of *Prevotella* and *Xylanibacter* spp., while a Western diet, which generally contains saturated fat and protein from animals, is associated with an increase in *Bacteroides* spp. and a decrease in general microbiota gene diversity [30]. Insoluble forms, such as cellulose, have a fecal-increasing effect and resist metabolism by the gut microbiota, especially in the unit host. The dietary fiber that microbes can use for carbon and energy includes microbe-accessible carbohydrates [148]. Given that the diet provides both soluble and insoluble fiber, while the most generally used diets contain only insoluble cellulose, an imbalance between fiber sources has a well-established impact on the gut microbiota and metabolism, which has been demonstrated in important experimental results [149].

Today, fat tissues are generally known as physiologic organs that act on the endocrine metabolism. Fatty tissues store excess nutrients in neutral lipids in an overnutrition state, while undernutrition activates lipolysis and fuels lipid decomposition [150]. From a nutritional point of view, a low intake of vegetables, fruits, protein, grains, and ω3-fatty acids, along with an inadequate and excessive intake of saturated fats and calorie oversupply, are major causes of NAFLD [151]. 

Dietary fructose is strongly associated with the pathogenesis of NAFLD. A large association study found that people who regularly consume sugary drinks, especially those who are obese, show an increased risk of NAFLD [152]. Dietary fructose may cause rapid and deleterious changes in the composition of the gut microbiota, including reduced phylogenetic diversity and low concentrations of the genus *Bifidobacterium* [153]. Fructose leads to gut bacterial imbalances that increase gut macrophages and reduced tight junction function, which cause translocation and endotoxemia, higher bacterial potential, and increased liver TLR expression [154]. The intake of artificial sweeteners is nowadays also associated with a dysbiosis of the gut microbiota.

Several types of foods have been linked to microbiota. Fermented foods contain a lot of substances beneficial to health and are used in various forms depending on the region. Beans (Cheonggukjang, doenjang, fermented bean curd, miso, natto, soy sauce, stinky tofu, tempeh, and oncom), grains (amazake, beer, bread, choujiu, gamju, injera, kvass, makgeolli, murri, ogi, rejuvelac, sake, sikhye, sourdough, sowans, and rice wine), vegetables (kimchi, mixed pickle, sauerkraut, Indian pickle, gundruk, and tursu), fruit (wine, vinegar, cider, perry, and brandy), honey, dairy, fish (bagoong, faseekh, fish sauce, garum, and ngapi), meat (chorizo, salami, sucuk, pepperoni, nem chua, and som moo), and tea are well-known fermented foods.

## 5. Conclusions

In recent years, gut microbiota have received great attention in chronic liver disease. The microbiota can enhance energy extraction from the diet, regulate systemic metabolism towards increased fatty acid uptake in adipose tissue, and shift the lipid metabolism from oxidation to de novo production. Therefore, a key role in maintaining gut–liver axis health is attributed to the gut microbiota. 

All foods are absorbed in the intestine and used as an energy source by organs, so foods we eat are closely related to the gut microflora. For healthy activity of the gut microbiota, a nutrient-balanced diet and regular meals are necessary. In addition, since heavy drinking, tobacco, drug abuse, lack of exercise, and lack of sleep affect the gut microflora, a healthy diet is necessary along with a healthy life habit.

The liver can be greatly affected by changes in gut microbiota through the portal vein of the liver, and the gut–liver axis is especially important for understanding the pathophysiology of various liver diseases. Gut microbiota-targeted therapy for liver disease is related to metabolites and disease mechanisms, and further research is required for targeted therapy of the gut microbiota (Figure 1). In addition, further studies for new diagnostic methods using specific gut microbiota are needed in the future.

## Figures and Tables

**Figure 1 foods-11-02703-f001:**
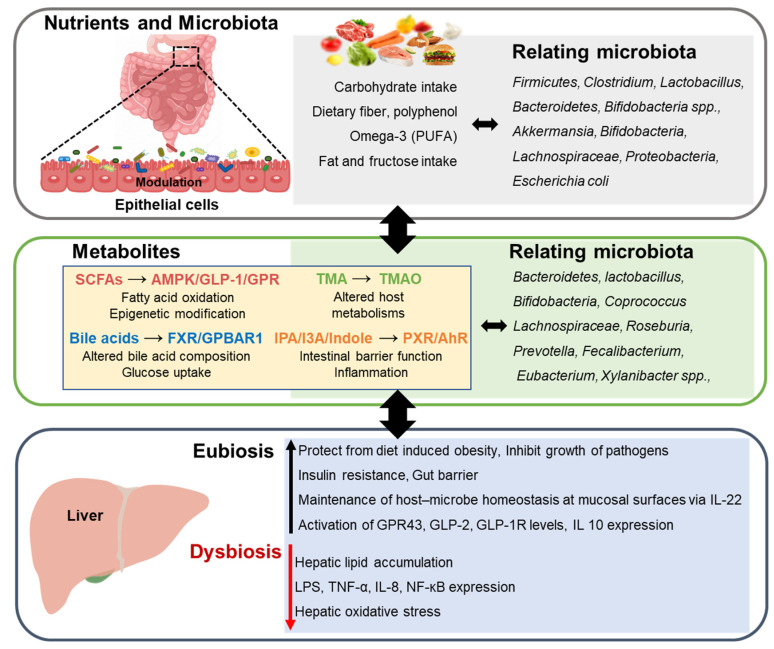
Interaction between nutrition and gut microbiome in the progression of chronic liver disease. SCFAs, short-chain fatty acids; GPR43, G-protein-coupled receptor 43; GLP, glucagon-like peptide; LPS, lipopolysaccharide; IL-10, interleukin-10; IL-22, interleukin-22; NF-κB, nuclear factor kappa-light-chain-enhancer of activated B cells; TNF, tumor necrosis factor; TMA, trimethylamine; TMAO, trimethylamine-N-oxide; IPA, indole-3-propionic acid; I3A, indole-3-acetic acid; PXR, pregnane X receptor; AhR, aryl hydrocarbon receptor; AMPK, AMP-activated protein kinase; GLP-1, glucagon-like peptide-1; GPR4, G-protein-coupled receptor; FXR, farnesoid X receptor.

**Table 1 foods-11-02703-t001:** Changes in gut microbiota and metabolites in different types of diets.

Nutrient	Microbiota Changes	Altered Metabolites	Reference
Protein	(↑): *Bacteroidetes, Lactobacillus*	(↑): Sulfide, polysaccharide lyases, tryptophan catabolism	[26,27]
	(↓): *Firmicutes, Clostridium*		
Omega-3	(↑): *Bifidobacteria, Lachnospiraceae, Roseburia, Bacteroidetes*	(↑): SCFAs	[28,29]
	(↓): *Enterobacteria, Faecalibacterium*	(↓): IL-1β, IL-6, TNF-α	
Fiber	(↑): *Prevotella, Xylanibacter* spp., *Bifidobacterium, Roseburia, Faecalibacterium*	(↑): SCFAs	[30,31,32]
Low-fiber	(↑): *Akkermansia, Bacteroides caccae*		[33]
	(↓): *Escherichia coli*		
Fat intake	(↑): *Firmicutes, Clostridium*	(↑): LPS, Indoxyl sulfate, p-cresyl sulfate	[34,35]
	(↓): *Lactobacillus, Bacteroidetes, Bifidobacteria* spp., *Akkermansia*		
Fructose intake	(↑): *Clostridium innocuum, Catenibacterium mitsuokai, Enterococcus* spp.	TMAO	[36,37]
High-fat diet	(↑): *Firmicutes, Proteobacteria*	(↑): TMAO, LPS	[24,38,39,40]
	(↓): Ba*cteroidetes, Bifidobacteria*		

(↑) increase; (↓) decrease. LPS, lipopolysaccharide; TMAO, trimethylamine N-oxide; SCFA, short-chain fatty acids; PUFA, polyunsaturated fatty acid; IL-1β, interleukin-1 beta; IL-6, interleukin-6; TNF, tumor necrosis factor.

**Table 2 foods-11-02703-t002:** The alterations in gut microbiota in the pathogenesis of nonalcoholic fatty liver disease.

Liver Disease	Microbiota Changes	Major Impacts	References
NAFLD	(↑): *Proteobacteria, Firmicutes, Lactobacillus, Parabacteroides, Allisonella, C. coccoides* (↓): *Oscillibacter, Faecalibacterium, Anaerosporobacter*	(↑): FFAs, triglycerides, de novo lipogenesis (↓): ApoB	[45,46,47]
NAFLD-associated Cirrhosis	(↑): *Streptococcus, Lactococcus, Enterobacteriaceae* (↓): *Bacteroidetes, Bacillus, F. prausnitzii, Prevotella*	(↑): PDGF, TGF-β, ECM	[48,49]
NAFLD-associated HCC	(↑): *E. coli*, *Actinobacteria* (↓): *Parabacteroides*, butyrate-producing genera	(↑): HBV, HCV, Wnt/β-catenin, JAK/STAT, Rb, p53, MAPK	[50,51]
**Model**			
Methionine choline-deficient diet model, leptin receptor deficiency steatosis model, and the high-fat diet model.	(↑): Hepatic steatosis, hepatic inflammation, lipid biosynthesis (↑): *Bacteroidetes, Prevotellaceae, Deferribacteres, Oscillibacter* (↓): *Lactobacillus, Bacteroidetes*		[52]
High-fat (45% energy) or low-fat (10% energy) diet for 10 weeks.	(↑): Body weight (by 34%) (↓): Hepatic steatosis, hepatic inflammation (↑): *L. gasseri, L.s taiwanensis*		[53]
High-fat diet-induced and lean mice (7 weeks): supplemented with *B. pseudocatenulatum*	(↓): Insulin resistance, hepatic fat, serum inflammatory markers, body weight (↑): *Bifidobacteria, Enterobacteriaceae*		[54]

(↑) increase; (↓) decrease. NAFLD, nonalcoholic fatty liver disease; HCC, hepatocellular carcinoma; ALD, alcohol-related liver disease; FFAs, free fatty acids; PDGF, platelet-derived growth factor; ECM, extracellular matrix; HBV, hepatitis B virus; HCV, hepatitis C virus; JAK, Janus kinase; STAT, signal transducer and activator of transcription; MAPK, mitogen-activated protein kinase; ROS, reactive oxygen species.

**Table 3 foods-11-02703-t003:** Nutrients associated with microbiota in nonalcoholic fatty liver disease.

Nutrients	Microbiota	Function	Reference
Bile acid	*Bifidobacterium, Lactobacillus, Enterobacter, Clostridium, Bacteroides*	(↑): Inhibit growth of pathogens, gut-derived hormone secretion, insulin resistance, gut barrier	[60,61,62,63]
SCFAs	*Bifidobacterium* sp., *Roseburia, Clostridium, Faecalibacterium, Coprococcus*	(↑): Protect from diet-induced obesity, (↓): Hepatic lipid accumulation	[64,65]
Acetate	*Bacteroidetes*	(↑): Insulin signaling, hepatic function, activation of GPR43 (↓): Fat accumulation, lipid storage	[66,67]
Propionate	*Bacteroidetes, B. obeum, C. catus, R. inulinivorans, P. copri*	(↑): Regulation of colonic T reg cell homeostasis,	
Butyrate	*Clostridia, F. prausnitzii, Eubacterium, Roseburia, C. catus, A. hadrus*	(↑): GLP-2, GLP-1R level, antiinflammation, gut–gut barrier function, T reg cell homeostasis, colonic suppression of colonic inflammation, activation of GPR43	[60,68,69]

(↑) increase; (↓) decrease. SCFAs, short-chain fatty acids; GPR43, G-protein-coupled receptor 43; GLP, glucagon-like peptide; LPS, lipopolysaccharide.

## Data Availability

Data are contained within the article.

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
