# Peer review of "Food and Gut Microbiota-Derived Metabolites in Nonalcoholic Fatty Liver Disease"

_foods, 2022, doi:10.3390/foods11172703_

Round 1

Reviewer 1 Report

This article covers an important topic that may be suitable for Foods. However, the quality of the English language is very poor. It gives the impression that the introductory sections and subsequent sections are written by different persons, as the quality of the English suddenly improves significantly. Therefore, I advise that the complete article is going to be rewritten and checked by a native English speaker before it is resubmitted to Foods for re-evaluation. 

Line 19: metabolisms à metabolites

·        Line 18-19: sentence unclear

·        Line 21: The microbiota (article)

·        Line 22: we discussed the influence of nutrients, gut microbes and their corresponding metabolites…

·        Line 31-32: Way too bold statement. A high calorie diet MAY cause metabolic syndrome..

·        Line 35: a lot of microbiotas: I guess the authors mean microorganisms.  “A lot of” is not scientific, please rephrase.

·        Line 37: an ecosystem that interacts with each other? Incorrect.

·        Line 41: Not “indicators”, but “drivers”

·        Line 42: microbial group à replace by “microbiota”.

·        Line 51 and 51: replace the sentence by:  Several studies have indicated that an inappropriate diet may lead to a fatty liver.

·        Line 58: microbiota à replace by “microorganisms”

·        Line 60-61 (each microbial..etc): unclear what is meant.

·        Line 62: remove “such as”

·        In the first paragraph the authors provide conflicting information. First the viruses in the microbiota, but later they define the microbiota as all organisms that live in the GI tract. Viruses are not living organisms. I plead for including them, but then your definitions have to be  updated and coherent. Als, the authors claim that 99% of the gut microbiota consists of bacteria. This is wrong. The number of bacteria is approximately 10EXP13 (see Sender et al.), and the number of viruses is estimated to be 10EXP14.

·        Line 69 till end of paragraph: refer more general to pathobionts with appropriate reference.

·        Line 74: “metabolisms” should be replaced by “metabolites”.

·        Line 74: “As a result” à Replace by: Intestinal dysbiosis is associated with various diseases such as…..”

·        Line 79: “Different people can have different gut microbiomes. This is wrong: the gut microbiota composition is unique for every individual.

·        Line 89: include reference with recent epidemiological data on metabolic syndrome and diabetes, which is put in the perspective of deterioration of the gut microbiota:

Larsen, Olaf FA, et al. "The Gut Microbiota: Master of Puppets Connecting the Epidemiology of Infectious, Autoimmune, and Metabolic Disease." Frontiers in Microbiology (2022): 1604.

·        Line 92: low intake of fats??

·        Line 152: Adapt sentence to: “Although most dietary……….small intestine, significant amounts….”

·        Line 172: please specify or give examples of what types of serious diseases.

·        Line 179-180: sentence is not correct.

·        Line 184: “Some gut microbiota”à I assume the authors mean “some gut microorganisms”?

·        Paragraph 4.4: a lot of articles (“the”)  are missing.

·        Line 206: explain abbreviation FXR

·        Paragraph lines 308 – 315: It is worthwhile mentioning that the intake of artificial sweeteners is nowadays also associated with a dysbiosis of the gut microbiota.

·        Line 348: explain the abbreviation HCC

·        Line 351: HCC incidence in the US is lower as compared to many countries. This is surprising. It would be interesting if the authors could provide an explanation for this.

·        Figure 1: it is from this figure not clear whether / how the bacterial taxa mentioned are associated to the processes (like SFA formation). If not, it is from this figure not clear why these specific taxa are being mentioned. 

Author Response

Comment 1: This article covers an important topic that may be suitable for Foods. However, the quality of the English language is very poor. It gives the impression that the introductory sections and subsequent sections are written by different persons, as the quality of the English suddenly improves significantly. Therefore, I advise that the complete article is going to be rewritten and checked by a native English speaker before it is resubmitted to Foods for re-evaluation. 

Reply: On behalf of my team, I convey my best gratitude to Reviewer 1 for his/her comments, which helped us to improvise this manuscript. We did English Editing service of MDPI. We marked change on track change marked version.

We are grateful to the reviewer for below valuable and reasonable comment.

Line 19: metabolisms à metabolites

We changed as reviewer suggested.

Line 18-19: sentence unclear:

     Recent advances in diagnostic technology for intestinal microbes and microbiota-derived metabolites

have led to advances in diagnosis, treatment, and prognosis of NAFLD.

Line 21: The microbiota (article)

We changed as reviewer suggested.

Line 22: we discussed the influence of nutrients, gut microbes and their corresponding metabolites…

We changed as reviewer suggested.

Line 31-32: Way too bold statement. A high calorie diet MAY cause metabolic syndrome. Line 35: a lot of microbiotas: I guess the authors mean microorganisms.  “A lot of” is not scientific, please rephrase.

We changed as reviewer suggested.

Line 37: an ecosystem that interacts with each other? Incorrect. Line 41: Not “indicators”, but “drivers”. Line 42: microbial group à replace by “microbiota”. Line 51 and 51: replace the sentence by:  Several studies have indicated that an inappropriate diet may lead to a fatty liver. Line 58: microbiota à replace by “microorganisms”

We changed as reviewer suggested.

Line 60-61 (each microbial..etc): unclear what is meant.

We erased unclear sentence.

Line 62: remove “such as”

     We removed such as

In the first paragraph the authors provide conflicting information. First the viruses in the microbiota, but later they define the microbiota as all organisms that live in the GI tract. Viruses are not living organisms. I plead for including them, but then your definitions have to be  updated and coherent. Als, the authors claim that 99% of the gut microbiota consists of bacteria. This is wrong. The number of bacteria is approximately 10EXP13 (see Sender et al.), and the number of viruses is estimated to be 10EXP14.

Line 69 till end of paragraph: refer more general to pathobionts with appropriate reference.

Reply: We are grateful to the reviewer for this valuable and reasonable comment. We changed the paragraph.

“There are many microorganisms in our intestines, which are collectively called the "gut microbiota" [13]. The gut microbiota comprises all organisms that present in the gas-trointestinal tract. Bacteria, viruses, fungi, and protozoa make up approximately 100 tril-lion microorganisms in the human gastrointestinal tract [14]. Each microbial species can form colonies from 1012~1014 cells/mL [15]. The composition and function of the gut micro-biota can be affected by various factors, including age, gender, genetic factors, lifestyle, medication, and dietary differences [16]. Environmental and genetic factors, as well as changes in the gut microbiota that play a role in the pathogenesis of metabolic disorders, are also implicated. Most gut bacteria exist as commensal form, including lactobacilli and bifidobacteria. However, there are also Enterococcus and Escherichia coli, which can be harm-ful under certain conditions such as dysbiosis or various diseases [17,18].”

Line 74: “metabolisms” should be replaced by “metabolites”. 74: “As a result” à Replace by: Intestinal dysbiosis is associated with various diseases such as…..”

We changed as reviewer suggested.

Line 79: “Different people can have different gut microbiomes. This is wrong: the gut microbiota composition is unique for every individual.

      We changed sentence. “Each individual harbors his own distinctive pattern of microbial composition, and chang-es in the diet lead to changes in the composition.”

Line 89: include reference with recent epidemiological data on metabolic syndrome and diabetes, which is put in the perspective of deterioration of the gut microbiota: Larsen, Olaf FA, et al. " Frontiers in Microbiology (2022): 1604.

      We inserted reference.

Line 92: low intake of fats??

      We changed “low” to “unbalanced”

Line 152: Adapt sentence to: “Although most dietary……….small intestine, significant amounts….”

We changed as reviewer suggested.

Line 172: please specify or give examples of what types of serious diseases.

We changed as reviewer suggested.

Line 179-180: sentence is not correct.

      We erased sentence.

Line 184: “Some gut microbiota”à I assume the authors mean “some gut microorganisms”?

We changed as reviewer suggested.

Paragraph 4.4: a lot of articles (“the”)  are missing. Line 206: explain abbreviation FXR. Paragraph lines 308 – 315: It is worthwhile mentioning that the intake of artificial sweeteners is nowadays also associated with a dysbiosis of the gut microbiota.

We changed as reviewer suggested.

Line 348: explain the abbreviation HCC. Line 351: HCC incidence in the US is lower as compared to many countries. This is surprising. It would be interesting if the authors could provide an explanation for this.

We added “Low rate of blood transmission, advances in treatment, and competing mortality risks among NAFLD patients are related with decrease of HCC incidence rates in US [167].”

Figure 1: it is from this figure not clear whether / how the bacterial taxa mentioned are associated to the processes (like SFA formation). If not, it is from this figure not clear why these specific taxa are being mentioned. 

We changed Fig 1.

Reviewer 2 Report

The author plans to write about the role of food and gut metabolites in non-alcoholic fatty liver disease or the role of gut microbiota in metabolizing food-produced metabolites in NAFLD, clearly stated? Moreover, the logic of the manuscript is confused, and much of the content is irrelevant to the subject. It is suggested that the author carefully revise and submit again. It can write about the role of gut microbiota in NAFLD-related dietary metabolites and the corresponding metabolites in NAFLD.

1. The introduction did not clarify the domestic and foreign research progress on NAFLD caused by intestinal microorganisms.

2. Three tables and Figure 1 appear in the manuscript, please indicate where the tables and Figure 1 are referenced?

3. LPS is a component of the outer wall of intestinal Gram-negative bacteria cell wall, which also plays an important role in the occurrence and development of NAFLD. It is suggested to be added to the manuscript.

4. The symbols of the references do not appear in the same font in the manuscript. For example, Line 186, [83,84]

5. The title of the manuscript, food and intestinal Metabolites in NAFLD, is ambiguous; Is it the gut microbiota that uses metabolites produced by food? Little is said in the manuscript about the role of food in NAFLD?

6. Line 200, 5. Microbial metabolism in non-alcoholic fatty liver disease

This paragraph does not correspond to the content, which is mainly about the relationship between some food ingredients and NAFLD. The previous section described four classes of intestinal metabolites, but little was said about the relationship between these four classes of metabolites and NAFLD. It's a good title. It should be written according to the title. Right? It turns out it wasn't

7. Line 322, 6. Liver cirrhosis

Are these parts rarely involved in the effects of food and gut microbes on liver cirrhosis? Since NAFLD is divided into three types, why only write one and it is not relevant to the topic?

8. Line 386, Fig. 2? Can't find Figure 2?

9. Gut-liver axis is mentioned in the abstract and conclusion of the paper, but it is rarely mentioned in the manuscript.

Author Response

The author plans to write about the role of food and gut metabolites in non-alcoholic fatty liver disease or the role of gut microbiota in metabolizing food-produced metabolites in NAFLD, clearly stated? Moreover, the logic of the manuscript is confused, and much of the content is irrelevant to the subject. It is suggested that the author carefully revise and submit again. It can write about the role of gut microbiota in NAFLD-related dietary metabolites and the corresponding metabolites in NAFLD.

Reply: On behalf of my team, I convey my best gratitude to the Reviewer 2 for his/her comments, which helped us to improvise this manuscript relativity to the research and readability  for better understanding of our work and findings in the scientific community.

  1. The introduction did not clarify the domestic and foreign research progress on NAFLD caused by intestinal microorganisms.

Reply: We appreciate the Reviewer’s thoughtful comment. We changed introduction and revised  it.

“In the intestine, nutrients are metabolized by intestinal microbiotas, and the metabo-lized nutrients and their metabolites move to the liver through the portal vein and are in-volved in various metabolic processes. This gut-liver axis is involved in the metabolism of nutrients and microbiota in the gut-liver axis plays an important role. The trillions of mi-crobes living in the gut may contribute to the pathogenesis of liver diseases [3]. The gut mi-crobiota affects liver disease through various mechanisms, including chronic systemic in-flammation, increased intestinal permeability, the production of short-chain fatty acids, and changes in the metabolism [4]. The gut-microbiome dysbiosis is related with NAFLD progression by increasing translocation of microbial metabolites into the liver [5,6].”

  1. Three tables and Figure 1 appear in the manuscript, please indicate where the tables and Figure 1 are referenced?

Reply: As reviewer suggested, we inserted referenced Figure and Tables.

  1. LPS is a component of the outer wall of intestinal Gram-negative bacteria cell wall, which also plays an important role in the occurrence and development of NAFLD. It is suggested to be added to the manuscript.

Reply: Thanks for this suggestion. We added suggested comment.

“Uncontrolled diet may cause dysbiosis and gut-derived microbial lipopolysaccharide (LPS), a toxic component of bacterial wall, play an important role in the development of NAFLD [6].”

  1. The symbols of the references do not appear in the same font in the manuscript. For example, Line 186, [83,84]

Reply: We checked font in the manuscript.

  1. The title of the manuscript, food and intestinal Metabolites in NAFLD, is ambiguous; Is it the gut microbiota that uses metabolites produced by food? Little is said in the manuscript about the role of food in NAFLD?

Reply: We appreciate the Reviewer’s thoughtful comment. We agree with reviewer’s opinion. We changed subject.

“Food and Gut Microbiota-derived Metabolites in Nonalcoholic Fatty Liver Disease”

  1. Line 200, 5. Microbial metabolism in non-alcoholic fatty liver disease

This paragraph does not correspond to the content, which is mainly about the relationship between some food ingredients and NAFLD. The previous section described four classes of intestinal metabolites, but little was said about the relationship between these four classes of metabolites and NAFLD. It's a good title. It should be written according to the title. Right? It turns out it wasn't

Reply: This is a very important point that we totally agree with. We realized this paragraph does not correspond to the content. So, we erased some paragraph and combined subject 4 and subject 5. In addition, we changed the location of Table and Figure. 

  1. Line 322, 6. Liver cirrhosis

Are these parts rarely involved in the effects of food and gut microbes on liver cirrhosis? Since NAFLD is divided into three types, why only write one and it is not relevant to the topic?

Reply: This is a very important point that we totally agree with. We erased all cirrhosis part.

  1. Line 386, Fig. 2? Can't find Figure 2?

Reply: We appreciate the reviewer’s correction of the mistake. We changed “2” to “1”

  1. Gut-liver axis is mentioned in the abstract and conclusion of the paper, but it is rarely mentioned in the manuscript.

Reply: We inserted paragraph in the manuscript.

“Gut-liver axis is important for the development, progression, and prognosis of NAFLD. According to the kind of diet and food, intestinal dysbiosis can occurs, and Bacteria de-rived substances [endotoxin, bacterial DNA, microbe-associated molecular patterns (MAMPs), damage-associated molecular patterns (DAMPs), pathogen-associated molecu-lar patterns (PAMPs), and nematode-associated molecular patterns (NAMPs)] directly af-fect the liver through the portal vein [6]. In addition, alcohol-secreting strains cause an in-flammatory response in the liver and body[59].”

Round 2

Reviewer 2 Report

The current manuscript is much improved, but a few minor problems still need to be corrected.

1. Gut microbiome, intestinal microbiotas, gut microbiota, gut microorganisms confusion in the manuscript to use these words, is there a difference?

2. There are some minor errors in the manuscript that need to be carefully corrected, such as, a unbalanced intake of fats, a or an ?  There is a blank paragraph at the bottom 4.6. ω-3 polyunsaturated fatty acid

3. The title was “4. Microbial metabolism in nonalcoholic fatty liver disease”, Choline, Polyphenol, ω-3 polyunsaturated fatty acid, Dietary fructose; These are not metabolites of the gut microbiota, right?

4.  In order to follow the title closely, it is suggested that the author add a summary and discussion of some foods, which can be used by intestinal microbiota to produce some metabolites mentioned in the manuscript, such as SCFAs, and have beneficial effects on NAFLD or adjuvant treatment.

Author Response

Revision for foods-1849580

“Food and Gut Microbial Metabolism in Non-alcoholic Fatty Liver Disease”

Reviewer 1

The current manuscript is much improved, but a few minor problems still need to be corrected.

Comment 1: Gut microbiome, intestinal microbiotas, gut microbiota, gut microorganisms confusion in the manuscript to use these words, is there a difference?

Reply: We appreciate the Reviewer’s thoughtful comment. Microbiome is the community of microorganisms that can usually be found living together in any given habitat. Microbiota is microorganism. Microbiota includes bacteria, archaea, protists, fungi, and viruses. To remove confusion, we changed “intestinal” to “gut” and “microorganism” to “microbiota”

  1. There are some minor errors in the manuscript that need to be carefully corrected, such as, a unbalanced intake of fats, a or an ? There is a blank paragraph at the bottom 4.6. ω-3 polyunsaturated fatty acid

Reply: We checked some minor errors and changed.

  1. The title was “4. Microbial metabolism in nonalcoholic fatty liver disease”, Choline, Polyphenol, ω-3 polyunsaturated fatty acid, Dietary fructose; These are not metabolites of the gut microbiota, right?

Reply: We are grateful to the reviewer for this valuable and reasonable comment. We changed title. “Nutrients associated with microbiota in nonalcoholic fatty liver disease”

  1. In order to follow the title closely, it is suggested that the author add a summary and discussion of some foods, which can be used by intestinal microbiota to produce some metabolites mentioned in the manuscript, such as SCFAs, and have beneficial effects on NAFLD or adjuvant treatment.

Reply: On behalf of my team, I convey my best gratitude to Reviewer 2 for comments, which helped us to improvise this manuscript. We added two paragraphs in the others and conclusion

“Several types of foods have been linked to microbiota. Fermented foods contain a lot of substances beneficial to health and is used in various forms depending on the region. Beans (Cheonggukjang, doenjang, fermented bean curd, miso, natto, soy sauce, stinky tofu, tempeh, and oncom), grains (amazake, beer, bread, choujiu, gamju, injera, kvass, makgeolli, murri, ogi, rejuvelac, sake, sikhye, sourdough, sowans, and rice wine), vegetables (kimchi, mixed pickle, sauerkraut, Indian pickle, gundruk, and tursu), fruit (wine, vinegar, cider, perry, and brandy), honey, dairy, fish (bagoong, faseekh, fish sauce, garum, and ngapi), meat (chorizo, salami, sucuk, pepperoni, nem chua, and som moo), and tea are well-known fermented foods.”

“All foods are absorbed in the intestine and used as an energy source by organs, so foods we eat are closely related to the gut microflora. For healthy activity of the gut micro-biota, a nutrient-balanced diet and regular meals are necessary. In addition, since heavy drinking, tobacco, drug abuse, lack of exercise, and lack of sleep affect the gut microflora, a healthy diet is necessary along with a healthy life habit.”
